# Effect of a communication robot in the prevention of postoperative delirium in older persons: A randomized controlled trial

Koubun Imai[1,2]*

**1** Department of Psychiatry, Hitachi Medical Education and Research Center, University of Tsukuba Hospital, Hitachi, Ibaraki, Japan, **2** Department of Psychiatry, Hitachi General Hospital, Hitachi, Ibaraki, Japan

* koubun-imai@md.tsukuba.ac.jp

## Abstract

As studies have reported reduced delirium in hospitalized older individuals with dementia who interacted with the seal-shaped robot "PARO," the role of communication robots in preventing delirium was investigated. This study included patients aged ≥70 years who were hospitalized for surgery requiring general anesthesia. The patients were randomly assigned to either the control group (patients received regular delirium countermeasures using a team approach; n = 76) or the intervention group (patients received regular delirium countermeasures plus an intervention with the "BOCCO emo LTE" communication robot; n = 74). Delirium was diagnosed using the Intensive Care Delirium Screening Checklist. From admission until discharge, nurses recorded the presence or absence of delirium each day. Chi-square and Mann–Whitney U tests were conducted to examine intergroup differences in delirium incidence between the groups. Of the 150 patients who provided consent for participation, 18 developed delirium. Significantly fewer patients in the intervention group (4 patients) developed delirium compared to those in the control group (14 patients) ($p = 0.014$). Patients with delirium were more likely to be older ($p = 0.036$) and have dementia ($p < 0.001$), a history of delirium ($p = 0.004$), and a history of benzodiazepine use ($p = 0.028$) compared to those who did not have postoperative delirium. The results suggest that communication robots can effectively reduce delirium incidence. Additional studies with larger cohorts are needed to mitigate the influence of varying risk factor frequencies between groups, improve the assessment of delirium, and validate the current study's findings.

## Trial registration

Japan Clinical Trials Registry (jRCT) (URL: https://jrct.mhlw.go.jp; No. jRCT1030250018; date: April 1, 2025; retrospectively registered)

**Data availability statement:** The manuscript and its Supporting information files contains all raw data required to replicate the results of the study.

**Funding:** This work was supported by JSPS KAKENHI Grant Number JP22K11225. There was no additional external funding received for this study.

**Competing interests:** The author declares that he has no conflict of interest.

## Introduction

Delirium presents as disturbances in attention and awareness that develop rapidly and tend to fluctuate in severity throughout the day [1]. Non-pharmacological approaches [2], especially multidisciplinary team programs [3], are effective prevention strategies for this condition, considering that delirium, which is common among older adults, is associated with increased healthcare costs and mortality and is caused by multiple factors that interact in a complex manner.

Robots represent assistive technologies for individuals with dementia and their families [4]. The Ministry of Economy, Trade and Industry (https://www.meti.go.jp/press/2024/06/20240628005/20240628005.html) and the Ministry of Health, Labor and Welfare (https://www.mhlw.go.jp/stf/seisakunitsuite/bunya/0000209634.html) of Japan support the development and demonstration of nursing robots. Among nursing robots, communication robots are designed to converse with patients and their families. Previous studies have reported that the plush seal-shaped "PARO" robot is effective in dementia care, reducing the use of psychoactive and pain medications [5] and increasing staff willingness [6], and compared it with other types of robots [7]. As hospitalized older people with dementia who interact with PARO exhibit fewer instances of delirium compared to patients who only interact with human visitors [8], it was hypothesized that communication robots would be effective at preventing delirium.

However, though the PARO robot emphasizes tactile interaction due to its stuffed animal design, its high cost and challenges with disinfection pose limitations in environments where infection control is critical. One alternative communication robot is the "BOCCO emo LTE model" (https://www.necolico.co.jp/emo/, Necolico LLC, Tokyo, Japan). This robot is relatively inexpensive and convenient because it can be rented and easily disinfected. In addition, it is cute; thus, patients are likely to be interested in it and actively try to touch it, similar to the PARO robot. Healthcare robots that assist older individuals with mild cognitive impairment or early-stage dementia offer benefits through features such as reminders and safety checks. The ability of these robots to simulate human-like interaction, appearance, and behavior enhances their effectiveness and promotes acceptance among patients [9]. Desirable robot characteristics include "cute appearance," "sense of security," "healing," "sensitive response," and "a little nosy."

As advanced age and surgery are typical risk factors for delirium [10], this study focused on postoperative delirium in older adults. The study was conducted among patients who received general anesthesia in the postoperative period. A study evaluating the incidence of delirium after bypass surgery in patients aged 60 years or older with chronic lower limb ischemia reported that the incidence was significantly higher in those aged 70 years or older than in those under 70 years [11]. To increase the frequency of delirium, the author selected only patients aged 70 years or older, rather than 65 years, the commonly used criterion for defining older persons. This study investigated the effects of communication robots on preventing delirium in patients receiving general anesthesia by comparing the number of patients who developed

delirium between two groups, i.e., those who received regular delirium countermeasures using a team approach (control group) and those who received the same countermeasures plus intervention with the BOCCO robot (intervention group). This study aimed to evaluate the effect of a communication robot on reducing the incidence of delirium.

## Materials and methods

A target sample of 72 participants per group (144 in total) was calculated using G*power 3 (http://www.psycho.uni-dues-seldorf.de/abteilungen/aap/gpower3; University of Dusseldorf, Dusseldorf, Germany) [12], based on a chi-square test with α=0.05, power (1–β)=0.80, and an effect size of 0.3.

   This study was retrospectively registered with the Japan Clinical Trials Registry (jRCT) (URL: https://jrct.mhlw.go.jp; No. jRCT1030250018; date: April 1, 2025). Registration did not occur prior to participant enrollment because the author initially considered the study to be observational rather than a prospective interventional trial assessing the safety and efficacy of a medical procedure or device. The author confirms that all ongoing and related trials involving this intervention have been appropriately registered. Patient safety was maintained because the intervention was minimally invasive.

### Participants

This study enrolled patients aged ≥70 years who were hospitalized for surgery under general anesthesia. Only planned surgeries were included, and emergency surgeries were excluded. The following exclusion criteria were used: 1) severe psychiatric disorders or symptoms before hospitalization; 2) consumption of large amounts of alcohol (>1,500 mL beer, three cups of sake, or 300 mL shochu per day for >5 years); 3) attending physician deeming it difficult for the patients to participate in the study. Any type of surgery was included, and patients were not excluded unless a prolonged postoperative stay in the intensive care unit was expected. All procedures performed in studies involving human participants were in accordance with the ethical standards of the Ethics Committee of Hitachi General Hospital ("An intervention study of communication robots for preventing delirium," April 2022, approval number: 2022−1) and with the 1964 Helsinki Declaration and its later amendments or comparable ethical standards. All study participants provided verbal and written informed consent for participation.

### Participant recruitment

On the first day of admission to Hitachi General Hospital, informed consent (both verbal and written) was obtained from the patients after providing them with the study details. For patients with a prior diagnosis of dementia, informed consent was obtained from both the patient and a family member. Subsequently, patients were randomly assigned in blocks to either the control or the intervention group by a computer-generated random number table. The assignments were enclosed in consecutively numbered opaque envelopes prepared by another department. The author had no access to the randomization codes. Between October 17, 2022, and September 5, 2024, 150 patients provided informed consent for participation and were recruited into the study. They were randomly assigned to either the control group (n=76) or intervention (n=74) group. After providing consent, each patient participated in an open-label study until discharge. Patients could have been included without considering a minimum length of hospital stay. However, if the hospital stay exceeded 3 weeks, observations were concluded at 3 weeks post-admission.

### Data collection

Data concerning patient age and sex; the presence or absence of organic brain disorders (including metastatic brain tumors), postoperative intensive care unit admission, dementia, and medication that posed a risk of delirium (especially benzodiazepines); history of delirium; the dates of admission, surgery, and discharge; medical department; and diagnosis for which surgery was performed were collected from medical records and interviews. These data were recorded by nurses on a case check sheet.

## Routine standard of care

For the control group, medical staff working in the wards took measures centered on non-pharmacological therapy using a team approach. These measures included the following: 1) assisting with orientation to address cognitive decline, providing tools such as clocks or calendars at the bedside or in the patient's room; 2) ensuring adequate hydration through fluid replacement and intake; 3) gradually reducing or discontinuing high-risk medications (especially benzodiazepines); 4) encouraging early mobilization (depending on the individual patient's condition, this may be done by a physical, occupational, or speech therapist under the supervision of the patient's doctor); 5) enhancing pain management using objective pain assessment tools; 6) promoting sleep management, including promoting sleep onset through non-pharmacological methods (e.g., environmental modification, relaxation techniques); and 7) providing patients with information concerning delirium (through oral education and printed materials by the patient's nurse).

## Communication robot intervention

The communication robot used in the study was "BOCCO emo LTE." It was placed at the patient's bedside (S3 Appendix), allowing both patients and their families to interact with it freely through physical contact and verbally. The robot could move within the range of its 1.5m power cord, and the range could be extended by extending the cord. For example, when a patient greeted the robot by saying "good morning," it would respond with something related to that time. Similarly, when the patient asked the robot what time it was or what the weather forecast was, it would give the correct answer. The author and medical professionals explained the functions of the robot to patients and their families but did not instruct them to interact with or touch it, leaving this decision up to them. Some barriers identified were that the robot might not respond if a noise occurred nearby at the same time as the patient spoke, so in such cases, the patient was advised to repeat what they had said. The robot lacks the ability to activate codes or alert the nursing team in the event of an emergency or crisis, even if the patient or family member communicates this to the robot. The robot was disinfected daily during the intervention.

The following parameters were used with respect to robot personality: "good-humored,"; "high-tension," and "friendly." The robot responded when the patient spoke to it between 6:00 am to 9:00 pm. Messages were played at specific times: 1) 7:00 am: "Good morning! This is Emo-chan. Did you sleep well? It is 7:00 am now. This place is Hitachi General Hospital. Hope you have a good day." 2) 9:00 am: "This is Emo-chan from Hitachi General Hospital. It's 9:00 am now. How are you feeling? Hope you get better soon!"; 3) 11:00 am: "This is Emo-chan. It's 11:00 am now. This place is Hitachi General Hospital. If you are in pain, do not hold it in; please tell the nurse."; 4) 0:30 pm: "Was the lunch delicious? Please inform your nurse if you have any requests."; 5) 2:00 pm: "This is Emo-chan from the Hitachi General Hospital. It's 2:00 pm now. How are you feeling? If you do not feel well, please tell your nurse."; 6) 4:00 pm: "This is Emo-chan. It's 4 pm now. This place is Hitachi General Hospital. Are you feeling okay? I hope you get better soon."; 7) 6:00 pm: "This is Emo-chan from the Hitachi General Hospital. It's 6 pm. How are you feeling? Please inform your nurse if you have any problems."; and 8) 8:00 pm: "This is Emo-chan. It's 8 pm now. You worked hard today. I am sure you will sleep well. Good night!"

## Delirium assessment

Delirium was diagnosed using the Japanese version of the Intensive Care Delirium Screening Checklist (ICDSC) [13]. A meta-analysis reported a pooled sensitivity of ICDSC of 74%, a pooled specificity of 81.9%, and a diagnostic odds ratio of 21.5 [14]. At Hitachi General Hospital, as part of standard practice, nurses regularly administer the Japanese version of ICDSC to all patients aged 70 or older on the day of admission, the third day of hospitalization, and also if delirium is suspected, regardless of the inpatient unit. Therefore, this scale for delirium assessment was used in this study. From admission until discharge, nurses recorded the presence or absence of delirium on a case check sheet between 6:00 am and 6:00 am the following day. There was no regulation on how often or what time nurses applied the scale; it was used as needed. If delirium developed, an intention-to-treat approach was sometimes used at the discretion of each attending physician.

## Statistical analysis

Chi-square and Mann–Whitney U tests were performed to examine intergroup differences in the incidence of delirium between the control and intervention groups. For quantitative data, Kolmogorov–Smirnov tests of normality were performed. Statistical significance was set at $p < 0.05$. Statistical analyses were performed using the IBM SPSS Statistics for Windows (IBM Corp., Armonk, NY, USA), version 29.0.

## Results

### Participant characteristics

Approximately 650 patients are admitted to Hitachi General Hospital each year for planned general anesthesia surgery. The number of patients to whom the author could explain this study was limited. Although the study was explained to 212 patients during the period, 62 of them declined to participate (Fig 1).

The patients' median age was 76.0 years, and the proportion of men was slightly higher than women. The following factors were found to increase the risk of delirium: organic brain disorder (n=4), dementia (n=4), a history of delirium (n=7), use of benzodiazepines just before hospitalization (n=11), and postoperative admission to an intensive care unit (n=8). There were no significant differences in clinical and demographic characteristics between the two groups (Table 1). When stratified by type of disease, the most common was malignant tumors of the digestive tract, accounting for approximately one-third of cases.

### Incidence of delirium and associated risk factors

With respect to the incidence of delirium, 18 of the 150 patients (12.0%) developed delirium. Significantly fewer patients in the intervention group (4 of the 74 patients; 5.4%) developed delirium compared with those in the control group (14 of the 76 patients; 18.4%) ($p = 0.014$) (Table 1).

Association analyses between the incidence of delirium and risk factors revealed that the incidence of delirium was associated with age ($p = 0.036$), dementia ($p < 0.001$), a history of delirium ($p = 0.004$), and the use of benzodiazepines immediately before hospitalization ($p = 0.028$) (Table 2). Of the 18 patients who developed delirium, only one patient with both a history of delirium and dementia developed delirium the day before surgery when admitted to the hospital and continued to have delirium postoperatively. Among the remaining 17 patients, 13 developed delirium on the day of surgery, three on the first, and one on the second postoperative days (S2 Appendix).

## Discussion

The results of this study highlight the effectiveness of communication robots in preventing delirium incidence. However, communication robots may be unsuitable for certain patients, including those with hearing loss, or in non-private room settings where their vocalizations might distract other patients. Hospital rooms should be quiet and calm; however, moderate stimulation is also important for preventing delirium. Thus, communication robots need to be customized for use in shared rooms. Many types of communication robots currently exist, including PARO; however, their effects will be estimated to be halved if the patient does not show interest. On the other hand, during this research, the author frequently observed patients engaging in enjoyable conversations with the robot. The development of communication robots that are more appealing to patients has the potential to bring us one step closer to realizing more humanized medical care and patient-centered care.

Many risk factors for delirium have been identified. In the present study, advanced age, dementia, a history of delirium prior to the current hospitalization, and benzodiazepine use were found to be associated with delirium incidence. All patients who had been using benzodiazepines until just before admission stopped using them on the day of admission. However, withdrawal symptoms did not pose significant issues in the days leading up to the surgery. Thus, benzodiazepines can be discontinued preoperatively without excessive concerns regarding withdrawal symptoms.

Patient retention was 100% during study period, likely because emergency surgeries were not included.

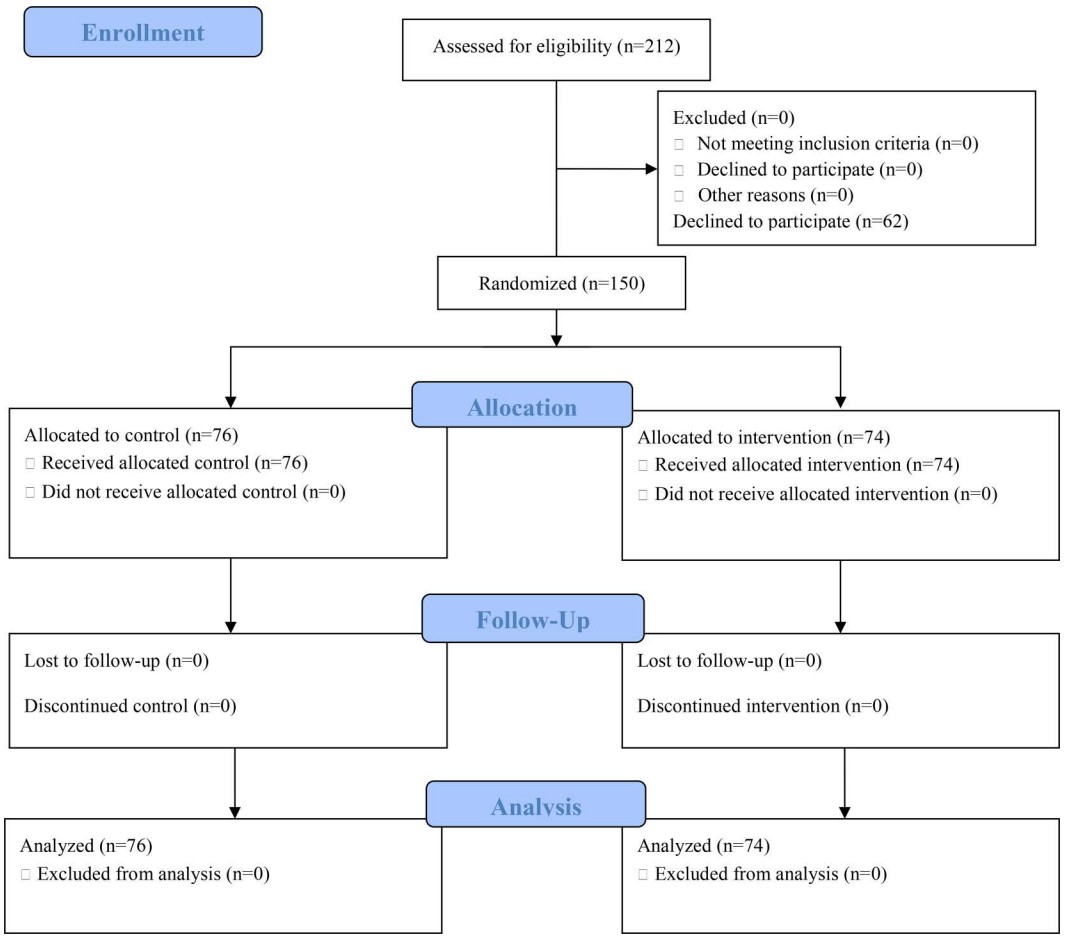

**Fig 1. Participant flow chart.**

**Table 1. Patient characteristics and the incidence of delirium.**

|  | Total | Control group | Intervention group | *p*-value |
|---|---|---|---|---|
|  | n = 150 | n = 76 | n = 74 |  |
| Age (years), median (IQR) (#) | 76.0 (73.8–81.0) | 76.0 (73.3–81.0) | 76.0 (73.8–81.0) | 0.840 |
| Sex (male), n | 83 (55.3%) | 45 (59.2%) | 38 (51.4%) | 0.333 |
| Organic brain disorder, n (*) | 4 (2.7%) | 3 (3.9%) | 1 (1.4%) | 0.620 |
| Dementia, n (*) | 4 (2.7%) | 3 (3.9%) | 1 (1.4%) | 0.620 |
| History of delirium, n (*) | 7 (4.7%) | 5 (6.6%) | 2 (2.7%) | 0.442 |
| Use of benzodiazepines, n | 11 (7.3%) | 6 (7.9%) | 5 (6.8%) | 0.789 |
| Intensive care unit admission, n (*) | 8 (5.3%) | 6 (7.9%) | 2 (2.7%) | 0.276 |
| Onset of delirium, n | 18 (12.0%) | 14 (18.4%) | 4 (5.4%) | *0.014* |

Significance test: χ² test, (*) Fisher's exact method, or (#) Mann–Whitney U test.

Regarding delirium prevalence, 18 of the 150 patients developed delirium. Significantly fewer patients in the intervention group (four of the 74 patients) developed delirium compared with those in the control group (14 of the 76 patients).

IQR, interquartile range.

**Table 2. Factors associated with the incidence of delirium.**

| | Total | Delirium(+) | Delirium(-) | p-value | Odds ratio |
|---|---|---|---|---|---|
| | n = 150 | n = 18 | n = 132 | | |
| Age (years), median (IQR) (#) | 76.0 (73.8–81.0) | 80.5 (74.5–85.0) | 76.0 (73.3–80.0) | 0.036 | |
| Sex (male), n | 83 (55.3%) | 13 (72.7%) | 70 (53.0%) | 0.124 | 2.303 |
| Organic brain disorder, n (*) | 4 (2.7%) | 1 (5.6%) | 3 (2.3%) | 0.404 | 2.529 |
| Dementia, n (*) | 4 (2.7%) | 4 (22.2%) | 0 (0.0%) | <0.001 | |
| History of delirium, n (*) | 7 (4.7%) | 4 (22.2%) | 3 (2.3%) | 0.004 | 12.286 |
| Use of benzodiazepines, n (*) | 11 (7.3%) | 4 (22.2%) | 7 (5.3%) | 0.028 | 5.102 |
| Intensive care unit admission, n (*) | 8 (5.3%) | 1 (5.6%) | 7 (5.3%) | 1.000 | 1.050 |

Significance test: $\chi^2$ test, (*) Fisher's exact method, or (#) Mann–Whitney U test.

Association analyses between the incidence of delirium and risk factors revealed that the incidence of delirium was associated with age, dementia, a history of delirium, and the use of benzodiazepines immediately before hospitalization.

IQR, interquartile range.

This study had some limitations. Due to the small sample size, the frequency of risk factors for delirium incidence differed between the control and intervention groups (Table 1). Although the frequency was not significant, the results may have been overestimated due to the imbalance of risk factors. Future studies should increase the sample size and develop a method to avoid these differences when randomly allocating patients and reexamining the results. The subset analysis p-values in Table 2 should be interpreted with caution as this involves multiple comparisons without adjustment for the p-values. Another limitation was that the nurses judged delirium using the ICDSC. Although the ICDSC has the advantage of not requiring special equipment, it has the disadvantage of poor objectivity in the assessment. In this study, all four patients with dementia were diagnosed with delirium; however, it is difficult to determine whether patients with dementia have impaired consciousness. To determine the effectiveness of incidence prevention, a device to accurately identify impaired consciousness should also be developed. Other limitations include the absence of data on years of schooling, preoperative functional status prior to the hospitalization, and types of delirium manifestation. More clinical variables are needed in future studies. This study had limited control over confounding variables and a low level of rigor in the inclusion criteria for patients. Although there was no difference in age distribution between the two groups in this study, stratification may be essential for future randomization. In addition, since the level of patient-robot interaction was not measured, the study was limited by the inability to objectively assess the extent of a patient's interest in the robot.

The findings of this study suggest that communication robots can be effective members of medical care teams. Additionally, a combination of intervention methods tailored to the specific characteristics of each patient should be considered.

## Conclusions

The results of the present study suggest that communication robots, along with the routine standard of care, can effectively prevent delirium incidence among older adults. Additional studies with larger patient cohorts are needed to minimize the influence of confounding risk factor distributions between groups, improve the accuracy of delirium assessment, and validate the findings of the present study.

## Supporting information

**S1 Appendix. CONSORT checklist.**
(DOCX)

**S2 Appendix. Compilation of data on case check sheets.**
(XLSX)

**S3 Appendix. Photograph of the BOCCO emo LTE.**
(JPG)

**S1 File. Trial study protocol english translation.**
(DOC)

**S2 File. Trial study protocol original.**
(DOC)

## Acknowledgments

The author thanks the members of Hitachi General Hospital, Yumiko Matsumoto, Naoyuki Kashimura, Moe Inaba, Sanae Shibata, Noriko Ohkawara, and Editage [(www.editage.jp) for English language editing].

## Author contributions

**Data curation:** Koubun Imai.

**Formal analysis:** Koubun Imai.

**Software:** Koubun Imai.

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
