## [Decision Letter · Decision Letter 0]

PONE-D-25-07404Effectiveness of a communication robot in the prevention of postoperative delirium: a randomized case control studyPLOS ONE

Dear Dr. Imai,

Thank you for submitting your manuscript to PLOS ONE. After careful consideration, we feel that it has merit but does not fully meet PLOS ONE’s publication criteria as it currently stands. Therefore, we invite you to submit a revised version of the manuscript that addresses the points raised during the review process.

We look forward to receiving your revised manuscript.

Kind regards,

Pedro Kallas Curiati, M.D., Ph.D.

Academic Editor

PLOS ONE

**Journal Requirements:**

1. When submitting your revision, we need you to address these additional requirements. Please ensure that your manuscript meets PLOS ONE's style requirements, including those for file naming. The PLOS ONE style templates can be found at https://journals.plos.org/plosone/s/file?id=wjVg/PLOSOne_formatting_sample_main_body.pdf and https://journals.plos.org/plosone/s/file?id=ba62/PLOSOne_formatting_sample_title_authors_affiliations.pdf 2. We note that you have selected “Clinical Trial” as your article type. PLOS ONE requires that all clinical trials are registered in an appropriate registry (the WHO list of approved registries is at https://www.who.int/clinical-trials-registry-platform/network/primary-registries"
https://www.who.int/clinical-trials-registry-platform/network/primary-registries and more information on trial registration is at http://www.icmje.org/about-icmje/faqs/clinical-trials-registration/). Please state the name of the registry and the registration number (e.g. ISRCTN or ClinicalTrials.gov) in the submission data and on the title page of your manuscript. a) Please provide the complete date range for participant recruitment and follow-up in the methods section of your manuscript. b) If you have not yet registered your trial in an appropriate registry, we now require you to do so and will need confirmation of the trial registry number before we can pass your paper to the next stage of review. Please include in the Methods section of your paper your reasons for not registering this study before enrolment of participants started. Please confirm that all related trials are registered by stating: “The authors confirm that all ongoing and related trials for this drug/intervention are registered”. Please see http://journals.plos.org/plosone/s/submission-guidelines#loc-clinical-trials for our policies on clinical trials. 3. Thank you for stating in your Funding Statement: This work was supported by JSPS KAKENHI Grant Number JP22K11225. Please provide an amended statement that declares *all* the funding or sources of support (whether external or internal to your organization) received during this study, as detailed online in our guide for authors at http://journals.plos.org/plosone/s/submit-now.  Please also include the statement “There was no additional external funding received for this study.” in your updated Funding Statement. Please include your amended Funding Statement within your cover letter. We will change the online submission form on your behalf. 4. We note that your Data Availability Statement is currently as follows: All relevant data are within the manuscript and its Supporting Information files. Please confirm at this time whether or not your submission contains all raw data required to replicate the results of your study. Authors must share the “minimal data set” for their submission. PLOS defines the minimal data set to consist of the data required to replicate all study findings reported in the article, as well as related metadata and methods (https://journals.plos.org/plosone/s/data-availability#loc-minimal-data-set-definition). For example, authors should submit the following data: - The values behind the means, standard deviations and other measures reported;- The values used to build graphs;- The points extracted from images for analysis. Authors do not need to submit their entire data set if only a portion of the data was used in the reported study. If your submission does not contain these data, please either upload them as Supporting Information files or deposit them to a stable, public repository and provide us with the relevant URLs, DOIs, or accession numbers. For a list of recommended repositories, please see https://journals.plos.org/plosone/s/recommended-repositories. If there are ethical or legal restrictions on sharing a de-identified data set, please explain them in detail (e.g., data contain potentially sensitive information, data are owned by a third-party organization, etc.) and who has imposed them (e.g., an ethics committee). Please also provide contact information for a data access committee, ethics committee, or other institutional body to which data requests may be sent. If data are owned by a third party, please indicate how others may request data access.

Reviewers' comments:

Reviewer's Responses to Questions

**Comments to the Author**

1. Is the manuscript technically sound, and do the data support the conclusions?

Reviewer #1: Yes

Reviewer #2: Partly

Reviewer #3: No

2. Has the statistical analysis been performed appropriately and rigorously? 

Reviewer #1: Yes

Reviewer #2: No

Reviewer #3: No

3. Have the authors made all data underlying the findings in their manuscript fully available?

Reviewer #1: Yes

Reviewer #2: Yes

Reviewer #3: Yes

4. Is the manuscript presented in an intelligible fashion and written in standard English?

Reviewer #1: Yes

Reviewer #2: Yes

Reviewer #3: Yes

5. Review Comments to the Author

**Reviewer #1: ** The study is relevant, innovative, and methodologically sound. However, it is not a case-control study as stated, but rather a randomized controlled trial. This needs to be addressed.

I would be interested to understand why the authors chose to include only patients aged 70 and above, rather than starting at 65 years, which is more commonly used as the threshold for defining older adults.

**Reviewer #2: ** The paper is very well written and the statistical limitations are recognized by the authors. The study is really a pilot effort. Clearly the prevalence of risk factors is very small. They truly need a larger, perhaps balanced, sample . More clinical and/or demographic variables are needed in the Table 1 beyond the supplement 2 data check sheets. Future randomization certainly needs stratification as learned from the current information.

The sample size for a single variable (delirium vs. no delirium) frequency outcome comparison is certainly accommodated by the Statistical section G-Power calculation with 72 per group and effect size =0.3 for chi-square and power =0.80. Some dropout anticipation should be considered as well. The Mann Whitney for other endpoints is certainly reasonable.

The overall conclusion was reasonable that ,’ The results of the present study may (my word) suggest that communication robots along with routine standard of care among older adults can effectively prevent delirium onset.’ An initial logistic model analysis (delirium vs. no delirium) would have been informative with risk factors in the model to at least learn of possible multivariate associations. As noted, this is primarily a pilot effort and the results are, at best, descriptive.

The limitations in the 'Discussion' should be expanded to alert the reader that the subset analysis p-values in Table 2 should be interpreted with caution as this involves multiple comparisons with no adjustment for the p-values.

**Reviewer #3: ** Title:

• In the title, the term “efficacy” is used, whereas throughout the article, including in the study objective, the term “effect” is employed. Considering that these are similar but distinct concepts, it should be defined which term best aligns with the study’s aim. I suggest choosing “effect,” as it more closely reflects what is being proposed. Please standardize a single concept throughout the entire document.

• I also recommend adding the target population of the study (elderly).

Introduction:

• Concisely and objectively cite the specific outcome related to the focus “PARO” in individuals with dementia.

• In the last paragraph, add that the study was conducted in patients in the postoperative period who received general anesthesia.

Materials and Methods

• The first paragraph has been removed (the objective should be maintained in the last paragraph of the introduction, as already present in the manuscript).

• This study employs comparative groups (intervention and control) that have been randomized. Therefore, I understand that the study design is not case-control but a controlled randomized clinical trial. Please clarify this in the Materials and Methods section and update the title accordingly.

• The sample size calculation should be clearly explained. Was it based on a previous study? Did it consider an estimated dropout rate?

• It is not clear whether the objective of assessing safety was maintained after the change in the study design. To make the text more concise and fluid, I suggest maintaining this information only in the paragraph from line 96 to line 99, concluding after the word “trial.”

Participants

• Considering that in developed countries a person is generally regarded as elderly starting at 65 years old, why was the cutoff age set at 70 years or older?

• Were only age and use of general anesthesia considered as inclusion criteria? Was a severity score, for example, taken into account? Could patients with neurological conditions, on mechanical ventilation, or with communication disorders be included?

• Please clarify whether all post-surgical destination units could be included and whether all types of surgery are eligible for inclusion.

Participant Recruitment

• Explicitly state that blinding was not employed and justify this choice, including how it constitutes a limitation of the study.

• In the last paragraph, lines 126/127, clarify whether patients could have been included without considering a minimum length of hospital stay.

Data Collection

• If feasible, collect data on years of schooling, preoperative functional status prior to the hospitalization that motivated the admission, and the type and duration of delirium manifestation. If this is not possible, include it as a limitation of the study.

Routine Standard of Care

• Improve the description of approaches for cognitive decline. Specify the content of this intervention, including activities or strategies used to maintain or stimulate cognitive functions.

• Describe the professionals involved in the early mobilization protocol (e.g., physiotherapist, occupational therapist, speech therapist, etc.), including the frequency and intensity of mobilization activities.

• Besides the pain assessment tool, specify the interventions used for pain management, including the name of the assessment tool cited in the text and additional pain management strategies (e.g., analgesic protocols, positioning, etc.).

• What non-pharmacological methods are employed to promote sleep hygiene and management? Specify the techniques or strategies used (e.g., environmental modification, sleep protocols, relaxation techniques).

• Who is responsible for delivering educational measures about delirium, and what is the content of this information? Specify the method used (e.g., verbal education, printed material, audiovisual aids).

• Indicate whether tools such as clocks and/or calendars are available in the bed/room to assist with orientation.

Communication Robot Intervention

• Suggest including a photograph of the “BOCCO emo LTE” within the context of hospitalization, in accordance with the journal’s policies regarding images and photographs.

• Specify that interaction could be carried out verbally and through physical contact.

• Clarify whether the researcher and/or healthcare professional instructed patients and/or family members to interact with the robot, and if so, how frequently this occurred.

• Explain whether there were any interaction barriers when the patient approached the robot, or if all responses were possible regardless of the patient's approach.

• In cases of urgency or emergency, could the robot, if verbally activated by the family or patient, play any role in activating codes or alerting the nursing team?

• Consider providing an example of how the robot could interact based on a patient-initiated interaction.

• Indicate whether the robot is stationary, fixed in a single location, or if it has mobility. Make this clear.

• Was the level of patient-robot interaction measured? If not, note this as a limitation of the study.

Delirium Assessment

• Why was the "Japanese version of the Intensive Care Delirium Screening Checklist" selected? The name suggests its applicability in ICU settings. Was it used regardless of the patient’s admission unit in the postoperative period?

• Include information on the reliability, validity, sensitivity, and specificity of this assessment tool for delirium diagnosis.

• Although nursing staff recorded the occurrence of delirium in a spreadsheet between 6:00 a.m. and 6:00 a.m., it is unclear how frequently the scale was applied during this period (i.e., how many times per day delirium was screened).

Statistical Analysis

• Both in the statistical analysis (line 175) and in the study objective (line 91), the author proposes to assess the effect of the robot on the incidence of delirium; however, the results are presented as prevalence (line 185).

It is understood that if the study aims to identify the occurrence of delirium over the entire hospitalization period, the incidence should be measured. Therefore, the way the results are presented needs to be adjusted accordingly. Although the “Delirium Assessment” section clearly states that the proposal is to identify the occurrence of delirium throughout hospitalization, in the Methods section (lines 87-88), the objective contradicts this, leading the reader to believe that the goal is to reduce the onset (which would relate to prevalence).

Please make these adjustments throughout the document to ensure consistency and align the results presentation with the stated objective. This lack of standardization was also observed in the discussion (lines 245-246) and the conclusion (line 255).

• It is necessary to clarify whether the intention-to-treat approach was used. If it was not, this should be explicitly mentioned as a limitation of the study.

• I suggest including an Odds Ratio analysis to evaluate the association between variables.

Participant Characteristics

• Begin the presentation of the results by describing the number of patients screened (which is not included in the flowchart and should also be added), those eligible for study inclusion, and those excluded.

• In the characteristics section, it is necessary to clearly state whether there were significant differences in clinical and demographic characteristics between the groups.

• It is not necessary to include the IQR and the absolute and relative frequencies (% estimates) in the text; these details should be left exclusively in the table.

Prevalence of Delirium and Associated Risk Factors

• As previously mentioned, if the objective is to assess the incidence, the results in this section should be adjusted accordingly.

• Once the incidence is measured, include the time taken for delirium onset in each group.

• It is not necessary to include the IQR or the absolute and relative frequencies (%). These details should be exclusively presented in the table.

• There are two repeated paragraphs between lines 186-188 and lines 200-202; they should be consolidated or removed.

• Add a column with Odds Ratios in Table 2 to evaluate the association between the variables.

• Include in the results which type of surgery was most frequent and whether the surgeries were elective or emergency.

Discussion

• Write a brief paragraph justifying how there were no patient losses throughout the trial.

• Lines 222 and 245: define whether the focus is on effect or effectiveness, as previously mentioned.

• Line 230: clarify whether the patient’s delirium history refers to the current hospitalization or another context/moment.

• A reflection could be included on whether humanization in healthcare and patient-centered care goals with therapies are feasible or not when using robotic systems.

• Add that this study had limited control over confounding variables and low rigor in the inclusion criteria for patients.

6. PLOS authors have the option to publish the peer review history of their article (what does this mean? ). If published, this will include your full peer review and any attached files.

**Do you want your identity to be public for this peer review?** For information about this choice, including consent withdrawal, please see our Privacy Policy .

Reviewer #1: **Yes: ** Christina May Moran de Brito

Reviewer #2: No

Reviewer #3: **Yes: ** Lorena de Toledo Montesanti

---

## [Author Response · Author response to Decision Letter 1]

17 Jun 2025

Response to Reviewers is uploaded under "Attach Files" tab.

---

## [Decision Letter · Decision Letter 1]

Effect of a communication robot in the prevention of postoperative delirium in older persons: A randomized controlled trial

PONE-D-25-07404R1

Dear Dr. Imai,

We’re pleased to inform you that your manuscript has been judged scientifically suitable for publication and will be formally accepted for publication once it meets all outstanding technical requirements.

Kind regards,

Pedro Kallas Curiati, M.D., Ph.D.

Academic Editor

PLOS ONE

Additional Editor Comments (optional):

Reviewers' comments:

Reviewer's Responses to Questions

**Comments to the Author**

1. If the authors have adequately addressed your comments raised in a previous round of review and you feel that this manuscript is now acceptable for publication, you may indicate that here to bypass the “Comments to the Author” section, enter your conflict of interest statement in the “Confidential to Editor” section, and submit your "Accept" recommendation.

Reviewer #2: All comments have been addressed

2. Is the manuscript technically sound, and do the data support the conclusions?

Reviewer #2: (No Response)

3. Has the statistical analysis been performed appropriately and rigorously? 

Reviewer #2: (No Response)

4. Have the authors made all data underlying the findings in their manuscript fully available?

Reviewer #2: (No Response)

5. Is the manuscript presented in an intelligible fashion and written in standard English?

Reviewer #2: (No Response)

6. Review Comments to the Author

Reviewer #2: (No Response)

7. PLOS authors have the option to publish the peer review history of their article (what does this mean? ). If published, this will include your full peer review and any attached files.

**Do you want your identity to be public for this peer review?** For information about this choice, including consent withdrawal, please see our Privacy Policy .

Reviewer #2: No

---

## [Editor Report · Acceptance letter]

PONE-D-25-07404R1

PLOS ONE

Dear Dr. Imai,

I'm pleased to inform you that your manuscript has been deemed suitable for publication in PLOS ONE. Congratulations! Your manuscript is now being handed over to our production team.

Kind regards,

on behalf of

Dr. Pedro Kallas Curiati

Academic Editor

PLOS ONE